# Analyzing the Asymmetric Effect of Renewable Energy Consumption on Environment in STIRPAT-Kaya-EKC Framework: A NARDL Approach for China

**DOI:** 10.3390/ijerph19127100

**Published:** 2022-06-09

**Authors:** Youxue Jiang, Zakia Batool, Syed Muhammad Faraz Raza, Mohammad Haseeb, Sajjad Ali, Syed Zain Ul Abidin

**Affiliations:** 1Jiangsu Vocational Institute of Commerce, Nanjing 211168, China; jyx145@163.com; 2Department of Economics, National University of Modern Languages (NUML), Islamabad 44000, Pakistan; zbatool@numl.edu.pk; 3China Institute of Development Strategy and Planning, Wuhan University, Wuhan 430072, China; haseeb.ecb@gmail.com (M.H.); syedzain@whu.edu.cn (S.Z.U.A.); 4Institute for Region and Urban-Rural Development, Wuhan University, Wuhan 430072, China; 5School of Economics, Quaid-e-Azam University, Islamabad 44000, Pakistan; sajjad3991ali@gmail.com

**Keywords:** renewable energy, population, technology, GDP growth, EKC, CO_2_ emissions, NARDL

## Abstract

This study aims to analyze the asymmetric relation between renewable energy consumption and CO_2_ emissions in China using the STIRPAT-Kaya-EKC framework. To delve into the asymmetric effect of renewable energy consumption on the environment, the non-linear ARDL model is used. The results of this study confirm the asymmetric impact of renewable energy on the environment in the long run as well as in the short run. However, the negative shocks to renewable energy have a greater detrimental influence on the environment than the benign effect due to the positive shock to renewable energy. Population growth affects the environment in the short run, whereas technology only affects environment quality in the long run. Moreover, the study supports the EKC theory in China. This research emphasizes that the administration can improve the economy’s lifespan by allocating substantial funds to establish legislation to maintain a clean environment by subsidizing renewable energy infrastructure and research and innovations for low-carbon projects.

## 1. Introduction

Economic progress is imperative for every country to bring ease to the lives of the people living within the country’s boundaries. However, increased economic activities and industrialization come along with the greater risk of environmental degradation as it may cause land, air, and water pollution. Air pollution is the most dangerous among pollution types. It spreads to a large area and needs to be cared for with proper regulation with the inclusion of chronological research evidence. The industrial sector of any developed country depends largely on fossil fuels for their energy needs, causing CO_2_ emissions that lead to environmental degradation, bringing different ecological and health problems for living beings. At the same time, the other disadvantages related to it are energy issues triggered by variations in oil prices, global warming, and acid rain. To achieve sustainable growth and development, many countries are making efforts to replace non-renewable energy sources with renewable ones mainly to deal with the problem of environmental degradation. Among the 17 SDGs, climate change and green growth are some of the most challenging tasks for policymakers in both developed and developing countries to achieve, as it needs reforms at a broader level [1]. However, in recent years, investment in renewable energy showed a scintillating rise over the years. According to a new estimate from Energy Transition Investment Trends 2022 published by Bloomberg NEF (BNEF), the total global investment in renewable energy hit $755 billion in 2021, showing an increase of 27% compared to the figures of 2020, within which renewable energy projects, for example, solar parks and wind farms, received the most investment in 2021, making a total investment of $366 billion. It is also observed that countries are trying to bring about a transition from fuel-based cars to electric cars. As per the International Energy Agency (IEA), worldwide electric car revenue has increased by 140% in the first quarter of 2021 as compared to 2020. China topped the ladder in the energy transition by selling electric cars, followed by Europe, whereas sales in the United States doubled. Electric car investment and renewable energy industries have driven up the global investment pattern toward low-carbon energy.

China, the world’s 3rd largest country in standings of land area with the largest population in the world, has emerged as an economic giant in recent years. Increased oil consumption is directly related to China’s sustained economic growth. China is using around 13% of the world’s oil per day, which is the second-largest consumption in the world after America. It is evident from the IEA 2021 that China is responsible for 33% of the CO_2_ emissions in the world, the most among any country, with 11.9 billion tonnes of CO_2_ emitted in 2021, which makes it challenging to achieve the targets of energy security, low-carbon atmosphere, and environmental preservation, all being hampered because of continued high economic expansion. To deal with these consequences, a comprehensive plan for environmental sustainability is required [2].

Although the Renewable Energy Law was passed in China in 2006 to promote the consumption and production of renewable energy, the Chinese economy has accounted for 44% of world oil demand growth since 2015. Excess capacity in oil refineries has made China among the world’s biggest net exporters of refined commodities. Another reason for China’s increase in demand for crude oil is that China’s oil demand is shifting its weight from industry to consumer-driven. On the other hand, the increase in domestic air transportation has boosted kerosene demand dramatically [3]. However, China’s yearly crude oil imports have declined by 5.4% in 2021, for the first time since 2001, after Beijing enforced refinery industry regulations to prevent excess local fuel generation. While on the other hand, analysts also credit the decrease in crude oil imports and decreased energy reliance to effective engagement by domestic energy companies in domestic oil field exploration and utilization in recent years, which has improved domestic energy independence. In 2021, China boosted its total energy shift spending by 60% from 2020, substantially solidifying its global leadership position in investment in the energy transition. Even if China is slowing down growth in oil demand, the baseline figures are now huge enough that even a slower yearly percentage rate of growth can still result in a considerable yearly absolute rise in oil demand volume.

Thus, keeping in view, the struggle of the Chinese economy is making energy transitions yet remains the top CO_2_ emitter in the world. At the same time, negotiation with increasing economic growth is also not easy. However, renewable energy consumption is continuously growing in China and will be an interesting factor in evaluating its long-term environmental sustainability influences. On the other hand, according to Eyuboglu and Uzar [4], a negative shock to the economy has a detrimental impact on renewable energy generation. Regarding this, the study attempts to evaluate the asymmetric impact of renewable energy consumption in China in the STIRPAT-Kaya-EKC framework. The findings of this study will help academicians and policymakers understand how renewable energy consumption affects the environment asymmetrically. Examining the data individually for positive and negative shocks to renewable energy consumption will help the policymakers develop more precise tactics when deciding on energy policy in China. Nevertheless, figures for carbon dioxide emissions, renewable energy consumption, and economic growth are incorporated in Figure 1, Figure 2 and Figure 3, which indicate the continuous increase in CO_2_ emissions, economic growth, and renewable energy consumption in China. However, it has been seen that there is much increase in CO_2_ secretions, economic growth, and renewable energy after the year 2000 in China. However, renewable energy is found to be more profound in its consumption in China after 2010. So, it will be noticeable to examine the long-run influences of economic growth, its desired level of growth, and renewable energy on CO_2_ emission in China. It will also be an interesting context to measure these factors’ positive and negative shocks on carbon emissions. Moreover, the current study’s findings are significant to the environmentalist in making policies that would help reduce CO_2_ emissions in China.

The rest of the paper is as follows: Section 2 and Section 3 include a literature review with the statement of the research gap and a theoretical and econometric model with the discussion of methodology, data, and sources. Section 4 discusses and interprets results, and research is concluded in Section 5 with policy recommendations.

This study significantly validates the STIRPAT-Kaya-EKC framework by analyzing the asymmetric relation between renewable energy consumption and CO_2_ emissions in China. The significant and negative impact of GDP squared on CO_2_ emissions confirms the inverted U shape of the EKC hypothesis in China. Furthermore, renewable energy’s positive shock helps to mitigate environmental degradation in China.

## 2. Literature Review

Economic growth and environmental degradation are being discussed widely across the research community. The increasing pollution, including emissions of reactive nitrogen, methane CO_2_, and ozone precursors caused by economic expansion and population growth, drew researchers and policymakers’ attention toward green growth. However, high levels of growth and economic development bring prosperity and enable the economy to focus on environmental objectives. In the early 1970s, researchers identified GDP and population as key factors of environmental quality. The idea of Environmental Impact–Population–Affluence–Technology (hereinafter IPAT) was introduced by Ehrlich and Holdren [5] in 1971. Later on, many studies used the concept of IPAT to explain environmental degradation [6,7,8,9,10,11,12].

Focusing on GDP as a contributor to environmental degradation, the theory of Environment–Kuznets–Curve (hereinafter EKC) explains that contamination intensifies during the preliminary stages of economic expansion until it hits a tipping point where pollution decreases as income per capita rises [13]. While examining CKC (carbon Kuznets curve) and its long-term relationship, Pao and Chen [14] found that the ingesting of fossils plays its part expressively in the deterioration of the environment. However, the use of renewables improves the eminence of the environment by reducing carbon emissions. Since energy conservation policies may impede economic growth, a higher proportion of clean energy is required for long-term sustainability. Because numerous technological advancements in several economic sectors are necessary for the early stages of economic expansion, a high degree of pollution is unavoidable. For the period 1980–2012, Zoundi [15] explored the link in accordance with CO_2_ emissions and renewable energy consumption (hereinafter REC) in 25 African nations and found a negative association between REC and carbon emissions. Naz et al. [16] inspected the association amongst GDP and CO_2_ emissions in Pakistan and identified a positive correlation between GDP and environmental degradation, thus rejecting the inverted U-shaped EKC hypothesis while arguing that REC considerably reduces CO_2_ releases. Nevertheless, the empirical findings of Hu et al. [17] revealed the EKC hypothesis and indicate that growing the proportion of renewable energy consumption adds value to carbon reduction; that is, raising the amount of renewable energy consumption results in decarbonization. Working on the same line, Sarkodie et al. [18], in their study, found that CO_2_ emissions and environmental degradation are reduced when production and renewable energy and income levels rise at the same time, while they argue that in both carbon emanations and degradation functions, the EKC hypothesis is supported. Bölük and Mert [19] further examined the EKC hypothesis in EU countries during the years 1990–2008 in terms of the relation between carbon emissions, GDP, and energy intake. They concluded that the European Union economies do not have an inverted U shape of EKC and also determined that energy derived from renewable sources reduces emissions of CO_2_ by 50% compared to energy derived from traditional sources.

In attempts to find the solution to protect the environment, many researchers explored the influence of energy use and energy transition on environmental deprivation in different regions of the world. Energy generated through fossil fuels and investment activities and development positively influence contamination levels in the environment. By employing cointegration analysis, Hasnisah et al. [20] exposed that renewable forms of energy negligibly influenced carbon secretions in thirteen Asian emerging nations. However, the study also explained that the rising ingestion of non-renewables and faster economic development offset the advantage of renewables. The study suggests that by better understanding the various variables impacting CO_2_ emissions, the nations may build a strategic strategy to slow environmental deterioration. In a similar vein, Nathaniel and Iheonu [21] explored the one-way causation between renewable and non-renewable energy forms and carbon secretions in Africa. Based on empirical estimations, they conclude that REC lowers carbon emissions while non-REC contributes to CO_2_ emissions. Lin and Moubarak [22] and Dong et al. [23] investigated the REC–growth–CO_2_ relationship in the case of China and found bidirectional causation running between REC and growth in the long run. However, no indication of a long-run or short-run causation involving emissions of CO_2_ and REC is realized.

A clean form of energy recognized as renewable energy is familiar and predictable for environmental protection. It has mostly been seen that this clean form of energy negatively influences CO_2_ releases and helps to ensure environmental cleanliness. Bekhet and Othman [24] argue that renewables are the most momentous factors to consider when enlightening environmental eminence. According to Payne [25], emissions of carbon dioxide had a favorable influence on the REC. Silva et al. [26] inspect the liaison flanked by REC, GDP, and carbon secretions and describe that increasing renewable energy reduces per capita carbon emanations. Moreover, Charfeddine and Kahia [27] evaluated the influence of REC on the CO_2_ level of the MENA region and revealed a significant impact of REC on CO_2_ reduction. In comparison, Nathaniel et al. [28] found that REC does not contribute to environmental quality in MENA countries. Excluding South Africa, Khattak et al. [29] observed the effectiveness of a clean form of energy use on the atmosphere in the BRICS economy, and their findings show that REC has negative and noteworthy long-term impacts on CO_2_, implying that CO_2_ emissions may be lowered by expanding renewable energy usage and forest area. Mohsin et al. [30] also established a similar result in twenty-five emerging Asian nations. Belaïd and Zrelli [31] and Sharif et al. [32] demonstrate that expanding renewables is a realistic option for tackling energy independence while also lowering carbon emissions to safeguard future generations’ environment. Hanif [33] and Acheampong et al. [34] scrutinize the impression of REC on the environment in Sub-Saharan African economies and discover that renewable energy sources promote air quality by reducing carbon emissions and reducing households’ direct exposure to toxic gases; thus, the usage of renewable aids economies in meeting their long-term development goals. Many researchers have conducted a comparative analysis of how REC affects the environment in low-income vs. high-income countries. In this vein, Nguyen and Kakinaka [35] illustrate that REC is positively related to carbon dioxide discharges in low-income economies, while the relationship is opposite for high-income nations. The empirical outcomes of Saidi and Omri [36] do not support a positive association amid economic progress and renewable energy consumption and CO_2_ emissions, demonstrating the validity of the neutrality hypothesis, which is explained by the uneven and inadequate utilization of renewable energy sources in the low-income countries. According to Gielen et al. [37], renewable energy can provide two-thirds of the world’s energy demand while also contributing to the reduction in greenhouse gas emissions. However, the increasing transition toward REC will help in controlling global warming [38]. Collender et al. [39] show that the risk premium on sovereign borrowing costs is lower in nations with lower carbon emissions, and advanced economies that fail to manage their climate transition may face higher sovereign borrowing rates. Lin and Zhu [40] consider CO_2_ emissions as an indicator of environmental change, investigate the responsiveness of renewable energy technology innovation to the intensiveness of CO_2_ releases in Chinese provinces, and conclude that the level of technical innovation varies significantly throughout China’s provinces, while at the same time, high CO_2_ emissions have boosted the amount of renewable energy technical innovation, implying that the innovation process is actively responding to climate change in China. Although there is fear that high energy prices caused by the rise of renewable energy would harm the economy, findings suggest that renewable energy use leads the economy toward sustainability [41]. On the other hand, Ali and Kirikkaleli [42] and Adebayo et al. [43] examine the asymmetric effect of renewable energy on CO_2_ emissions in Italy and Sweden, respectively.

Although the past studies in the literature have investigated the impact of renewable energy on environmental degradation, the proponents of energy transition argue that renewable energy is a benign form of energy. In contrast, other researchers have shown an insignificant inspiration for renewables in improving the environment. There are also studies that support the EKC theory, while others have rejected the EKC hypothesis. The literature discusses the asymmetric effects of energy consumption on the environment. However, the literature lacks providing the asymmetric effects of the environment influencing factors in the STIRPAT-Kaya-EKC framework for China. In this uncertain and unclear situation, this study utilizes the STIRPAT-Kaya-EKC framework analysis of the asymmetric effect of EKC in China.

## 3. Data, Model and Methodology

### 3.1. Theoretical Framework

According to classical economists, with increasing population density, the sustainable output cannot be maintained because the population grows geometrically while subsistence increases arithmetically. Thus, given the fixed resources, environmental degradation occurs if population growth exceeds the region’s carrying capacity. Later on, Ehrlich and Holdren [5] proposed an IPAT framework that highlighted the affluent and technology along with the population as potential drivers of environment quality. However, by upgrading the IPAT model, the existing research applies the stochastic influence through a regression on population growth, affluence, and technology (STIRPAT), which is acknowledged by Dietz and Rosa [44]. Thus, the empirical model can be written as follows:(1)$$ED_{t}\;=\beta_{0}+\beta_{1}P_{t}+\beta_{2}A_{t}+\beta_{3}T_{t}+e_{t}$$
where *ED* is environmental degradation measured by CO_2_ emissions, *P* is population growth, *A* is affluence proxied by per capita GDP, *T* is technology proxied by a number of patents, and *e* is the disturbance term. However, Grossman and Krueger [13] established the EKC theory that shows the link between income per capita and environmental degradation to prove an inverted U shape. Encompassing the EKC premise in Equation (1), the model takes the following form:(2)$$ED_{t}\;=\beta_{0}+\beta_{1}P_{t}+\beta_{2}GDP_{t}+\beta_{3}GDP_{t}^{2}+\beta_{4}T_{t}+e_{t}$$

The model in Equation (2) is further modified by adding the Kaya identity, which considers energy use as one of the determinants of environmental degradation. Since this study intends to evaluate the asymmetric impact of renewable energy on the environment, this study considers renewable energy use as one of the determinants. Thus, the model of this study can be written as
(3)$$ED_{t}\;=\beta_{0}+\beta_{1}P_{t}+\beta_{2}GDP_{t}+\beta_{3}GDP_{t}^{2}+\beta_{4}T_{t}+\beta_{5}RE_{t}+e_{t}$$
where the *RE* abbreviation is considered as renewable energy. To estimate the asymmetric influence of the consumption of renewable energy on the environment, this study collects annual data for the economy of China spanning the period 1990 to 2020. To capture environmental degradation, per capita CO_2_ emissions are used, and population data on the total population from WDI are used. To measure affluence, data on per capita real GDP are used, and to gauge the technological level, the number of resident patent applications is obtained. Data on renewable energy consumption are obtained from WDI. All the variables in Equation (3) are transformed in the logarithmic form, which helps avoid the problem of heteroscedasticity and produces efficient estimates.

### 3.2. Econometric Strategy

In order to evaluate the asymmetric effect of renewable energy consumption on environmental quality, this study employs the NARDL approach introduced by Shin et al. [45]. The NARDL version of the model given in Equation (3) can be written as follows:(4)$$\Delta ED_{t}=\theta_{0}+\sum_{i=1}^{p}\theta_{1i}\Delta ED_{t-i}+\sum_{i=0}^{q}\theta_{2i}\Delta GDP_{t-i}+\sum_{i=0}^{r}\theta_{3i}\Delta GDP^{2}{}_{t-i}+\;\sum_{i=0}^{s}\theta_{4i}\Delta P_{t-i}+\sum_{i=0}^{t}\theta_{5i}\Delta T_{t-i}+\sum_{i=0}^{u}\theta_{6i}^{-}\Delta RE_{t-i}^{-}+\sum_{i=0}^{u}\theta_{7i}^{+}\Delta RE_{t-i}^{+}\;+\pi_{1}ED_{t-1}\;+\;\pi_{2}GDP_{t-1}+\pi_{3}GDP_{t-1}^{2}+\pi_{4}P_{t-1}+\pi_{5}T_{t-1}+\pi_{6}^{-}RE_{t-1}^{-}+\pi_{7}^{+}RE_{t-1}^{+}+\mu_{t}$$

In Equation (4), *θ_i_* are the short-run coefficients, whereas *π_i_* are the long-run coefficients. The positive and negative sign over the renewable energy variable symbolizes the asymmetric effect of renewable energy on CO_2_ emissions. Following the approach used by Shin et al. [45] and Allen and McAleer [46], the positive and negative attributes of renewable energy are formulated using Equations (5) and (6), respectively.
(5)$$RE^{+}=\sum_{i=1}^{t}\Delta RE_{i}^{+}=\sum_{i=1}^{t}\max\;\left(\Delta RE_{i},0\right)$$
(6)$$RE^{-}=\sum_{i=1}^{t}\Delta RE_{i}^{-}=\sum_{i=1}^{t}\min\;\left(\Delta RE_{i},0\right)$$
where (*RE*)^+^ shows the partial sum of positive changes in renewable energy and (*RE*)^−^ is the partial sum of negative changes in renewable energy.

To assess the presence of a long-run association between dependent and independent factors, the bound technique is applied with the null hypothesis of no long-run equilibrium relationship, that is, H0: *π*_1_ = *π*_2_ = *π*_3_ = *π*_4_ = *π*_5_ = *π*_6_ = *π*_7_ = 0. Furthermore, the short-run and long-run asymmetries are examined by means of the Wald test with the null hypothesis of H0 = *θ_i_*^−^ = *θ_i_*^+^ = *θ* and H0 = *π*^−^ = *π*^+^ = *π*, respectively. The long-run estimates are derived from Equation (4), assuming that the differenced variables are zero, and normalizing the equation, we obtain the following model:(7)$${\mathrm{ED}}_{t}=\delta_{0}+\delta_{1}GDP_{t}+\delta_{2}GDP_{t}^{2}+\delta_{3}P_{t}+\delta_{4}T_{t}+\delta_{5}RE_{t}^{-}+\delta_{6}RE_{t}^{+}+e_{t}$$
where $\delta_{1}=$ $\pi_{2}/\pi_{1}$, $\delta_{2}=\pi_{3}/\pi_{1}$, $\delta_{3}=\pi_{4}/\pi_{1}$, $\delta_{4}=\pi_{5}/\pi_{1}$, $\delta_{5}=\pi_{6}^{-}/\pi_{1}$, $\delta_{6}=\pi_{6}^{+}/\pi_{1}$.

The short-run coefficient estimates of the independent variables are obtained using the restricted error correction method of the overhead model, which is given in Equation (4).
(8)$$\Delta ED_{t}=\theta_{0}+\sum_{i=1}^{p}\theta_{1i}\Delta ED_{t-i}+\sum_{i=0}^{q}\theta_{2i}\Delta GDP_{t-i}+\sum_{i=0}^{r}\theta_{3i}\Delta GDP^{2}{}_{t-i}+\;\sum_{i=0}^{s}\theta_{4i}\Delta P_{t-i}+\sum_{i=0}^{t}\theta_{5i}\Delta T_{t-i}+\sum_{i=0}^{u}\theta_{6i}^{-}\Delta RE_{t-i}^{-}+\sum_{i=0}^{u}\theta_{7i}^{+}\Delta RE_{t-i}^{+}\;+\pi_{1}ED_{t-1}\;+\;\lambda \;{\mathrm{ECT}}_{t-1}+\mu_{t}$$
where λ is the coefficient of the speed of adjustment and ECT is the error correction term obtained by the following equation.
(9)$${\mathrm{ECT}}_{t-1}={\mathrm{ED}}_{t}{-}_{1}-\delta_{0}+\delta_{1}GDP_{t-1}+\delta_{2}GDP_{t-1}^{2}+\delta_{3}P_{t-1}+\delta_{4}T_{t-1}+\delta_{5}RE_{t-1}^{-}+\delta_{6}RE_{t-1}^{+}$$

The application of the NARDL approach involves various steps. Firstly, the unit root test is applied to verify that there is no indication of spurious results which arises from variables integration at order two. Secondly, appropriate lag lengths for each of the variables are selected based on the Akaike information criterion. Thirdly, the occurrence of cointegration is examined using F-statistics, and if cointegration appears, then the asymmetric effects are calculated. Lastly, diagnostic assessments are practical to confirm the validity of the model, and QARDL is applied for a robustness check.

## 4. Results and Discussion

Before applying estimation techniques, the descriptive analysis of the variables in Table 1 is completed to understand the behavior of the data. It provides a picture of how data are distributed and assists with the detection of outliers and mistakes. The value of skewness shows that, except for technology, the distribution of other variables is symmetric.

### 4.1. Unit Root Test

In the first estimation step, this existing study employs the Augmented Dickey–Fuller and Phillips–Perron test to assess variables’ integration order. The null hypothesis of both tests states that there is a unit root against the alternative hypothesis of the stationary data set. The test statistic values are then compared with critical standards produced at significance levels 1, 5, and 10%. The unit root test results are reported in Table 2.

The results in Table 2 depict that the indicators are not stationary at levels, and all are integrated at order one. Since no variables are stationary in the second-order, we can carry on with the NARDL model.

### 4.2. Cointegration Analysis

To test the presence of a long run equilibrium association flanked by the variables, a NARDL bound test approach anticipated by Pesaran et al. [47] is employed. The result of the cointegration test in the NARDL framework is given in Table 3. The F statistics are equated with the critical standards produced by Narayan [48]. Narayan [48] argued that the critical standards developed by Pesaran et al. [47] are centered on large sample sizes. Therefore, they are not appropriate for small sample sizes. The bound F-test statistic is 5.573, which is greater than the critical value of the upper bound and extensive at a 5% level of significance. Thus, it is determined that there is a non-linear long-run equilibrium association between the model variables.

### 4.3. NARDL Model Results

After the validation of cointegration for the NARDL model, the subsequent step involves the examination of long-run and short-run asymmetries for renewable energy. The Wald test is applied to verify the asymmetry in renewable energy, and the test outcomes are stated in Table 4. Since the probability value of both short-run and long-run F-statistics in Table 4 is less than 0.05, we decline the null hypothesis of symmetric effects, which allows us to analyze the effect of negative and positive shocks to renewable energy on the carbon secretions.

The long-run and short-run evaluations of the NARDL model are given in Table 5. The short-run estimates in Panel A show that the value of negative shocks to renewable energy is significantly positive, indicating that a 1% decline in the REC affects the CO_2_ emissions by 1.29%. On the other hand, environmental quality improves due to positive shocks to renewable energy consumption. The coefficient value of positive shocks to renewable energy consumption is −0.506; that is, a 1% rise in the consumption of renewable energy causes CO_2_ emissions to reduce by 0.506%. However, rendering to the results, the negative shocks to renewable energy consumption are found to deteriorate the environment quality more than the positive effects of positive shocks to renewable energy consumption. The existing research result is in accordance with Rehman et al. [49]. According to the EKC theory, the linear term of GDP is expected to increase environmental degradation, while the quadratic term affects environmental degradation negatively. According to our outcomes, the coefficient of GDP is positive, and the square of GDP is negative, which validates the existence of EKC in the short run.

An increase in population is also found to contribute to environmental degradation as its coefficient value is positive; that is, a 1% surge in the population causes CO_2_ emissions to intensify by 0.722%. Many researchers have identified population size as one of the key drivers of environmental degradation and argued that populated economies are more likely to have high CO_2_ emissions [50,51]. The growing population results in a rise in aggregate demand, including demand for petroleum for transportation, electricity, and industrial goods [52,53]. According to the results, the number of patent applications which is used as a proxy for technology is not found to affect environmental degradation over the short run. The coefficient of ECT is −0.548, which is significant at a 5% level. The negative value of the adjustment coefficient shows that 54.8% of disequilibrium is corrected annually.

The estimates of long-run coefficients are given in Panel B of Table 5. The results show that the negative and positive shocks to renewable energy affect CO_2_ emissions significantly, correspondingly at 5% and 10% significance levels. The negative shock to renewable energy deteriorates the quality of the environment drastically; that is, a 1% decline in the intake of renewable energy drives CO_2_ emissions to increase by 2.897% in the long run. However, a 1% positive shock to renewable energy causes CO_2_ emanations to reduce by 0.762%. Regarding the effect of per capita GDP, the value of the linear term is 2.314 and significant in the long run, which is more or less similar to the short-run coefficient of linear GDP. On the other hand, the quadratic term is negative, and compared to the short-run coefficient of the quadratic term, GDP^2^ affects CO_2_ emissions to a lesser extent in the long run. The coefficient of linear and squared GDP shows that the EKC theory holds in the long run as well, which shows that at initial levels of growth, the environment deteriorates due to the scale effect. However, at high levels of GDP, the growth process becomes slower, and the pollution control initiatives can offset the scale effect. The coefficient of population growth is insignificant; thus, population growth affects environmental efficiency negatively in the short run. However, the population does not affect the environment in the long run in China. These outcomes of the existing research are in agreement with the studies of Ahmad et al. [54] and Shah et al. [55]. The technology originated to mark CO_2_ emissions negatively in the long run only; that is, a 1% increase in technology mitigates CO_2_ releases by 1.205%. Technological advancements in both agriculture and industry, such as the use of mechanization and sophisticated chemicals in agriculture, as well as the excessive use of technological machinery in the industrial sector, increase CO_2_ emissions. In contrast, the development of high-tech industries induces technologies that use a higher proportion of renewable energy over the long run, thus affecting CO_2_ emissions negatively [56]. Our result shows that in the case of China, due to renewable energy-bound technology, the negative consequence of technological advancements on CO_2_ emanations prevails over the long run.

Panel C of Table 5 illustrates the diagnostic analysis. The value of adjusted R^2^ is 0.687, which shows that around 69% of the variation in CO_2_ emissions is explained by the independent indicators of the model. The significant rate of the F-statistic indicates the overall significance, whereas the Portmanteau test up to lag 10 shows that there is no serial correlation. The probability rate of the J-B assessment is higher than 0.05, which leads to the conclusion that errors are normally distributed. At the same time, the insignificant value of RESET shows that the existing empirical model is appropriately specified. After this, we have examined the robustness test to assess the sensitivity of the findings. In Table 6 of QARDL, the findings illustrate that the value of GDP^2^ is significant at quantile v = 0.25, 0.50, and 0.75, and its negative sign validates the EKC inverted U-shape hypothesis. Furthermore, renewable energy is found to be confident in reducing environmental degradation. However, the findings of QARDL fulfilled the need for robustness, and there is no issue in the model. The findings are given below in Table 6.

### 4.4. Stability Graph Results

In the end, stability graphs of CUSUM and CUSUM-square are employed to witness the stability of the empirical model by graphs. Regarding this, two stability graphs, CUSUM and CUSUM-squared, are employed in which the CUSUM and CUSUM-square lines are prerequisites for positioned between the upper and lower bound lines. However, here in the results, CUSUM and CUSUM-square lines are found between the upper-lower bounds and presented in Figure 4 and Figure 5, evidencing the stability of the empirical model to measure STIRPAT-Kaya-EKC in China.

## 5. Conclusions and Policy Recommendations

This study employs the STIRPAT-Kaya-EKC framework and examines whether renewable energy affects environmental degradation asymmetrically in China or not. Following the empirical results, the main outcomes of the study are as follows:
The NARDL estimates of linear and quadratic terms of GDP support the EKC hypothesis in China in both the short-run and long-run, implying that the advantages of economic expansion in standings of CO_2_ secretions reduction will be realized over time.According to the findings, renewable energy has an asymmetric impact on environmental quality. However, results indicate that the negative shocks to renewable energy use have a greater damaging impression on the environment than the positive shock to renewable energy consumption. The decrease in the use of REC leaves the population to rely on non-renewable energy sources that cause a tremendous increase in CO_2_ emissions.It is also observed that the population contributes to environmental deprivation in the short run due to the resulting increase in demand for industrial products and transportation. However, its effect on environmental deterioration is insignificant in the long run.Technology proxied by the number of patent applications is found to improve the environment by shrinking the carbon releases in the long run, because the high-tech industries use energy-efficient and renewable energy-based technologies.

Based on empirical findings, this study emphasizes that the government can improve the economy’s lifespan by allocating substantial funds for environmental preservation and establishing legislation to maintain a clean environment. With the increase in economic expansion, the government of China must subsidize the clean energy projects to encourage the industrialists to use clean energy resources. According to our result, technology is found to affect environmental quality positively. Therefore, this study suggests that the government should encourage research and innovations in energy-saving and low-carbon projects. Since the findings of this study show that negative shocks to renewable energy consumption harshly harm the environment, the government needs to maintain subsidies for the use of renewable energy to avoid negative shocks to REC and improve the infrastructure of renewable energy production. Examining positive and negative shocks to REC would aid policymakers in establishing more precise methods when deciding on China’s energy strategy.

Future studies in this domain can investigate the asymmetric impact of the sub-components of renewable energy, for example, wind, hydro, solar, etc., on the environment. Furthermore, the upcoming research may focus on the asymmetric effect of sub-components of renewable energy on other populated and polluted countries.

## Figures and Tables

**Figure 1 ijerph-19-07100-f001:**
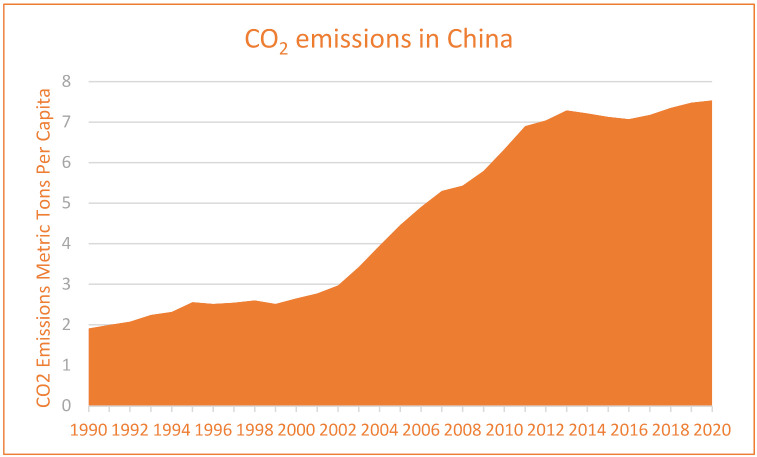
CO_2_ Emissions in China (1990–2020).

**Figure 2 ijerph-19-07100-f002:**
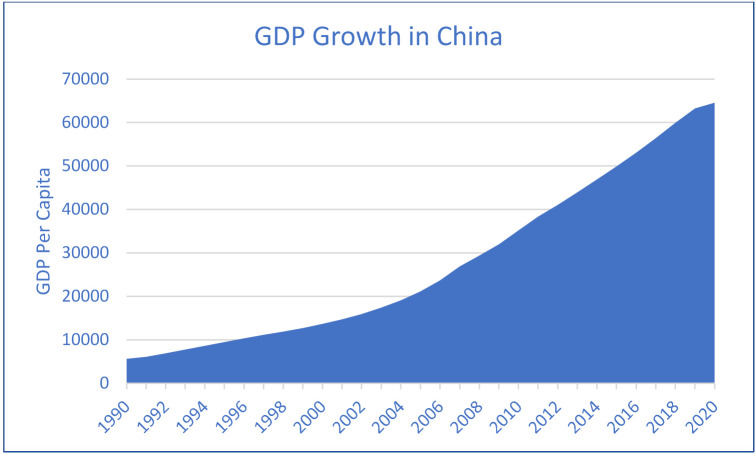
Economic Growth in China (1990–2020).

**Figure 3 ijerph-19-07100-f003:**
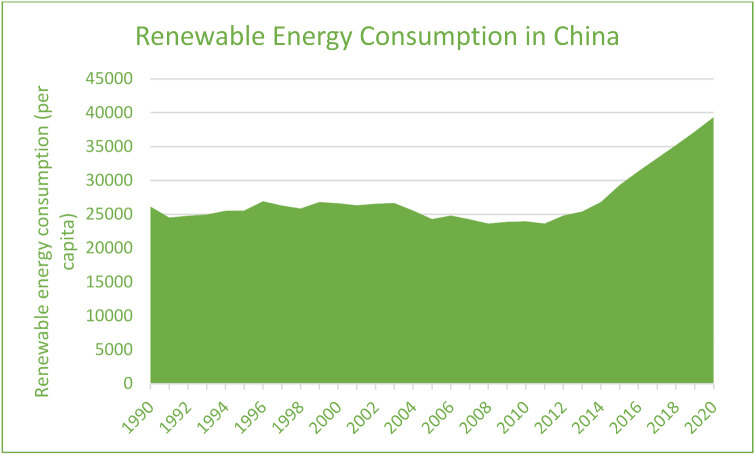
Renewable Energy Consumption in China (1990–2020).

**Figure 4 ijerph-19-07100-f004:**
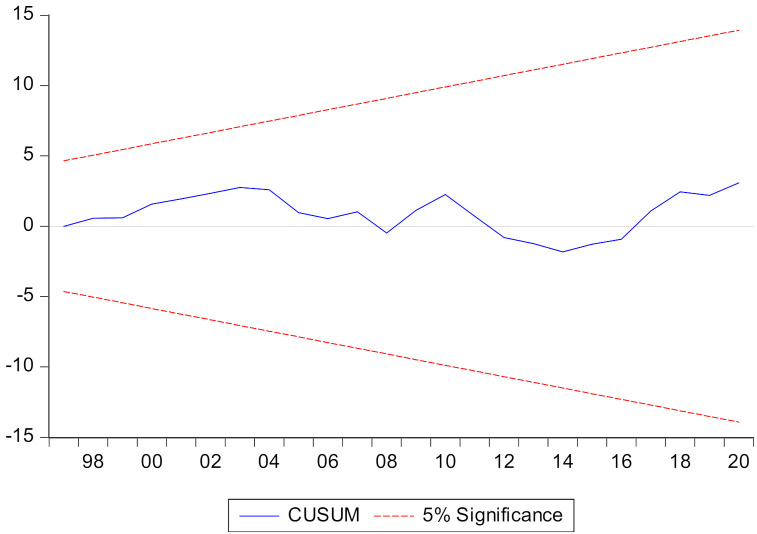
CUSUM.

**Figure 5 ijerph-19-07100-f005:**
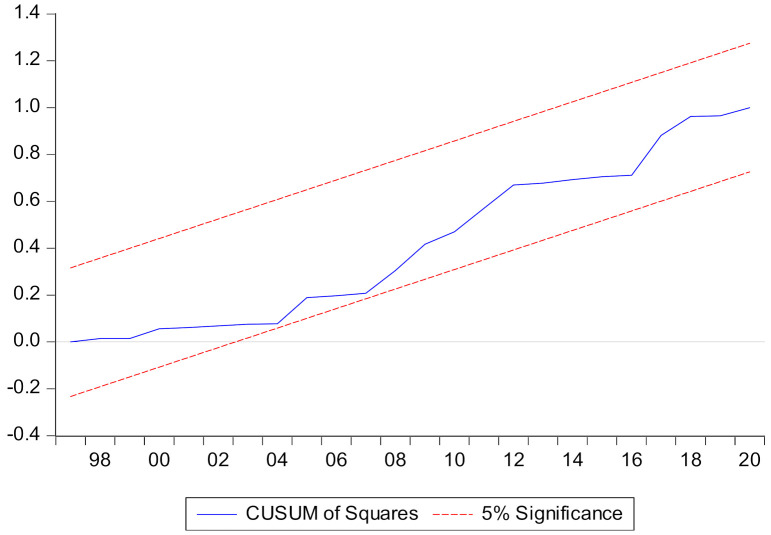
CUSUM of Squares.

**Table 1 ijerph-19-07100-t001:** Descriptive Statistics.

	ED	REC	GDP	T	P
MEAN	4.628	21.144	27,631.479	363,389.1	1.29 × 10^9^
S.D.	2.147	8.679	19,062.425	481,685.4	81,136,228
MIN	1.914	11.338	5636.079	5832	1.14 × 10^9^
MAX	7.538	34.083	64,581.412	1,393,815	1.41 × 10^9^
SKEW	0.134	0.212	0.618	1.173	−0.343
KURT	1.228	1.165	2.010	2.807	2.088

**Table 2 ijerph-19-07100-t002:** Results of Unit Root Tests.

Variable	Augmented Dickey-Fuller Test		Phillips-Perron Test	
	Level	1st Difference	Level	1st Difference
ED	−2.012	−3.328 *	−1.447	−3.531 **
GDP	−2.857	−4.529 ***	−2.034	−3.498 **
GDP^2^	−1.836	−3.894 **	−1.723	−3.277 *
P	−0.463	−4.552 ***	−0.439	−4.430 ***
T	−1.264	−3.291 *	0.012	−3.016 *
RE	−0.344	−4.967 ***	−0.773	−5.309 ***

Note: *, **, and *** show statistical significance at 10%, 5%, and 1% levels.

**Table 3 ijerph-19-07100-t003:** Result of NARDL Bound Test.

F-Statistics (PSS)	5.792	
Critical-Values		
Significance Levels	I (0)	I (1)
10%	2.977	4.260
5%	3.576	5.065
1%	5.046	6.930

Note: Narayan [48] is the source to acquire critical values for the bound test in accordance with k = 6 and *N* = 30 with no trend and unrestricted intercept.

**Table 4 ijerph-19-07100-t004:** Short-run and Long-run Asymmetry Statistics.

Short-Run Asymmetry		
Variable	F-Statistic	*p*-Value
RE	9.764 ***	0.008
**Long-run Asymmetry**		
RE	17.098 ***	0.000

Note: *** shows that values are meaningful at a 1% significance level.

**Table 5 ijerph-19-07100-t005:** The Short-run and Long-run Estimates of the NARDL Model.

Dependent Variable: ED		
Variables	Coefficient Value	*p*-Value
**Panel A: Short-Run Estimates**		
D (RE_NEG)	−0.660	0.118
D (RE_NEG (−1))	0.007	0.192
D (RE_NEG (−2))	1.291 ***	0.016
D (RE_POS)	−0.283	0.355
D (RE_POS (−1))	−0.506 *	0.092
D (GDP)	2.155 ***	0.001
D (GDP^2^)	−1.893 ***	0.008
D (P)	0.003	0.235
D (P (−1))	0.722 *	0.067
D (ED)	0.4776 **	0.034
ECT	−0.548 **	0.017
Constant	0.0118	0.776
**Panel B: Long-Run Estimates**		
RE_NEG	2.897 **	0.041
RE_POS	−0.762 *	0.081
GDP	2.314 ***	0.000
GDP^2^	−0.809 *	0.086
P	0.068	0.215
T	−1.205 **	0.013
**Panel C: Diagnostic Test**		
Adjusted R^2^	0.687	
F-Stat (*p*-value)	18.390 *** (0.000)	
Portmanteau test (10)	1.347 (0.265)	
JB Stat	3.722 (0.144)	
Ramsey RESET	1.794 (0.187)	

Note: *, **, and *** illustrate significance correspondingly at 10%, 5%, and 1%.

**Table 6 ijerph-19-07100-t006:** Estimates of QARDL Model.

Parameters	Quantile v = 0.25		Quantile v = 0.50		Quantile v = 0.75	
	Coefficient	*p*-Value	Coefficient	*p*-Value	Coefficient	*p*-Value
**GDP**	0.113	0.746	0.299	0.289	0.099	0.762
**GDP^2^**	−0.913 ***	0.000	−1.046 ***	0.000	−0.990 ***	0.000
**P**	0.014	0.842	0.031	0.357	0.000	0.980
**RE**	−0.692 ***	0.000	−0.526 ***	0.000	−0.535 ***	0.001
**T**	−1.328	0.681	−4.155	0.149	−4.227	0.144

Note: *** indicates the significance level at 1 percent.

## Data Availability

Data can be available on reasonable request.

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
