# Peer review of "Analyzing the Asymmetric Effect of Renewable Energy Consumption on Environment in STIRPAT-Kaya-EKC Framework: A NARDL Approach for China"

_ijerph, 2022, doi:10.3390/ijerph19127100_

Round 1
Reviewer 1 Report
This study is very interesting with a positive contribution. The authors are examining the asymmetric relation between renewable energy consumption and CO2 emissions in China using the STIRPAT-Kaya-EKC framework. However, I believe that some minor modifications are needed, before it is suitable for publication.
- Please cite the references according to IJERPH
- Lines 33-34: “Air pollution…..types” : why is air pollution the most dangerous pollution type? Please analyse and associate references.
- Lines 34-35: “it spreads ….inclusion of chronological research evidence” Please articulate about the inclusion of chronological research evidence and justify it based on literature.
- Lines 91-94: I suggest to move the mentioned figures into an appendix.
Author Response
Dear Reviewer,
We are truly thankful for your positive reply and for granting us the opportunity to improve the quality of our manuscript. The reviewer's comments are beneficial and enable us to address the minor revision to clarify the well-knowledgeable reviewers. Keeping the reviewers’ comments and suggestions in mind, we have edited the draft. We believe that the manuscript is substantially improved after making the suggested edits. Our reply to the comments of the reviewer is as under.
Comments and Suggestions for Authors
This study is very interesting with a positive contribution. The authors are examining the asymmetric relation between renewable energy consumption and CO2 emissions in China using the STIRPAT-Kaya-EKC framework. However, I believe that some minor modifications are needed, before it is suitable for publication.
Please cite the references according to IJERPH
Reply: Following the suggestion, we have cited the references according to IJERPH.
Lines 33-34: “Air pollution…..types” : why is air pollution the most dangerous pollution type? Please analyse and associate references.
Reply: Following the suggestion, we have analyzed and associated the reference as per the demand.
Lines 34-35: “it spreads ….inclusion of chronological research evidence” Please articulate about the inclusion of chronological research evidence and justify it based on literature.
Reply: Following the suggestion, we have justify the statement based on literature.
Lines 91-94: I suggest to move the mentioned figures into an appendix.
Reply: Thanks a lot for the suggestion. The figures are part of the discussion and placed in the introduction section to maintain the flow of the discussion. That is why we have placed them in the introduction section. Thanks for understanding this.

Reviewer 2 Report
The paper investigates the short-run and the long-run asymmetric impact of renewable energy consumption on carbon dioxide emissions in China. The NARDL framework has been employed to capture these effects. The technical analysis is appropriate and thorough. Suggestions to improve the quality of the contributions and the readability of the paper follow.
1) It is apparent that there is an increase in renewable energy consumption in China (and worldwide). The study aims to gauge the effects of positive and negative shocks of the renewable energy consumption on CO2 emissions. Under which circumstances negative renewable energy consumption shocks exist? Why these negative renewable energy consumption shocks matter?
2) Line 108: This paragraph provides the structure of the paper. It is customary to not mention what section 1/introduction covers. Furthermore, summarizing the key results in one or two paragraphs in the introduction would be valuable and effective on underscoring the contributions of the paper.
3) Page 5: There is literature review demonstrating the impact of renewable energy transition on CO2 emissions. Useful references such as Gielen et al. (2019) and Jones et al. (2016) provide a comprehensive assessment on this. Also, the importance of renewable energy consumption and CO2 on the climate change transition risk has been studied by Collender et al. (2021).
4) Line 217: The significance of the asymmetry in relation to the literature has not been motivated adequately.
5) Line 258: The model has been introduced by Shin et al. (2014) and Allen et al. (2021). These important relevant references should be added here.
6) Discuss the need to apply trend adjustments in the models. The impact of trend should be tested.
7) Table of descriptive statistics of the variables is not provided.
8) There are no robustness tests to assess sensitivity of findings. For example, a quartile regression can further quantify these asymmetric effects within quartiles.
9) The policy implications and the interpretations of the findings should be significantly refined. Furthermore, reflections on why the negative shocks tend to have higher impact compared to the positive shocks are essential and it would be very useful on highlighting the importance of the findings.
Allen, D., McAleer, M., 2021. A nonlinear autoregressive distributed lag (NARDL) analysis of the FTSE and S&P500 indexes, Risks, 9(11), 195.
Gielen, D., Boshell, F., Saygin, D., Bazilian, M.D., Wagner, N., Gorini, R., 2019. The role of renewable energy in the global energy transformation. Energy Strategy Review 24, 38-50.
Jones, G.A., Warner, K.J., 2016. The 21st century population-energy-climate nexus. Energy
Policy 93, 206-212.
Collender, S. and Nikitopoulos, S. C. and Richards, K.A. and Ryan, L. S., Climate Change Transition Risk on Sovereign Bond Markets (October 25, 2021). Available at SSRN http://dx.doi.org/10.2139/ssrn.3861350
Shin, Yongcheol, Byungchul Yu, and Matthew Greenwood-Nimmo. 2014. Modelling asymmetric cointegration and dynamic multipliers in a nonlinear ARDL framework. In The Festschrift in Honor of Peter Schmidt.: Econometric Methods and Applications. Edited by Robin C. Sickles and William C. Horrace. New York: Springer, pp. 281–314.
Author Response
Dear Reviewer,
We are truly thankful for your positive reply and for granting us the opportunity to improve the quality of our manuscript. The reviewer's comments are beneficial and enable us to address the major revision to clarify the well-knowledgeable reviewers. Keeping in mind the reviewers’ comments and suggestions, we have largely edited the draft while making revisions, including improving readability, re-framing motivation, and clarifying crucial spots. We believe that the manuscript is substantially improved after making the suggested edits. Our reply to the comments of the reviewer is as under.
Comments and Suggestions for Authors
The paper investigates the short-run and the long-run asymmetric impact of renewable energy consumption on carbon dioxide emissions in China. The NARDL framework has been employed to capture these effects. The technical analysis is appropriate and thorough. Suggestions to improve the quality of the contributions and the readability of the paper follow.
1) It is apparent that there is an increase in renewable energy consumption in China (and worldwide). The study aims to gauge the effects of positive and negative shocks of the renewable energy consumption on CO2 emissions. Under which circumstances negative renewable energy consumption shocks exist? Why these negative renewable energy consumption shocks matter?
Reply: Following the suggestion, we have cleared the statements, “Under which circumstances negative renewable energy consumption shocks exist? Why these negative renewable energy consumption shocks matter?”
2) Line 108: This paragraph provides the structure of the paper. It is customary to not mention what section 1/introduction covers. Furthermore, summarizing the key results in one or two paragraphs in the introduction would be valuable and effective on underscoring the contributions of the paper.
Reply: Following the suggestion, we removed the line covering section 1/introduction. However, the contribution of the paper by highlighting the key results is included in the introduction paragraph.
3) Page 5: There is literature review demonstrating the impact of renewable energy transition on CO2 emissions. Useful references such as Gielen et al. (2019) and Jones et al. (2016) provide a comprehensive assessment on this. Also, the importance of renewable energy consumption and CO2 on the climate change transition risk has been studied by Collender et al. (2021).
Reply: Following the suggestion, we have added these valuable studies in the literature, and the reference section is also updated with these references.
4) Line 217: The significance of the asymmetry in relation to the literature has not been motivated adequately.
Reply: Following the suggestion, the significance of the asymmetry is adequately added in the literature.
5) Line 258: The model has been introduced by Shin et al. (2014) and Allen et al. (2021). These important relevant references should be added here.
Reply: Following this valuable suggestion, we have added these references in the model discussion and reference section.
6) Discuss the need to apply trend adjustments in the models. The impact of trend should be tested.
Reply: Following the suggestion, the trend adjustments graph is included in the result section.
7) Table of descriptive statistics of the variables is not provided.
Reply: Following the suggestion, the descriptive statistic summary is incorporated in the result section.
8) There are no robustness tests to assess sensitivity of findings. For example, a quartile regression can further quantify these asymmetric effects within quartiles.
Reply: Following the suggestion, we have incorporated table 6 of QARDL to measure robustness in the result section.
9) The policy implications and the interpretations of the findings should be significantly refined. Furthermore, reflections on why the negative shocks tend to have higher impact compared to the positive shocks are essential and it would be very useful on highlighting the importance of the findings.
Reply: Following this valuable suggestion, we have refined the conclusion and policy implications section.
References
Allen, D., McAleer, M., 2021. A nonlinear autoregressive distributed lag (NARDL) analysis of the FTSE and S&P500 indexes, Risks, 9(11), 195.
Gielen, D., Boshell, F., Saygin, D., Bazilian, M.D., Wagner, N., Gorini, R., 2019. The role of renewable energy in the global energy transformation. Energy Strategy Review 24, 38-50.
Jones, G.A., Warner, K.J., 2016. The 21st century population-energy-climate nexus. Energy
Policy 93, 206-212.
Collender, S. and Nikitopoulos, S. C. and Richards, K.A. and Ryan, L. S., Climate Change Transition Risk on Sovereign Bond Markets (October 25, 2021). Available at SSRN http://dx.doi.org/10.2139/ssrn.3861350
Shin, Yongcheol, Byungchul Yu, and Matthew Greenwood-Nimmo. 2014. Modelling asymmetric cointegration and dynamic multipliers in a nonlinear ARDL framework. In The Festschrift in Honor of Peter Schmidt.: Econometric Methods and Applications. Edited by Robin C. Sickles and William C. Horrace. New York: Springer, pp. 281–314.
Reply: Thanks for providing us with the reference for these valuable studies. However, we have incorporated all these references in the reference section.
Round 2
Reviewer 2 Report
Most of the comments have been addressed.
The comment about the trend treatment has been misinterpreted. The questions on hand is about the need for "trend adjustment" in the models. The authors have only plotted the trend of each series, without any reflections on what it is observed and how it may affect the analysis. This is not useful thus it is recommended to remove Graph 1, unless they reflect on the impact of the trend on the models. Is a trend adjustment required for these models?
Author Response
Response Letter
Journal: International Journal of Environmental Research and Public Health
Manuscript ID: ijerph-1727966
Title: Analyzing the Asymmetric Effect of Renewable Energy Consumption on Environment in STIRPAT-Kaya-EKC Framework: A NARDL Approach for China
The authors are thankful to the reviewers for their constructive comments on the paper entitled “Analyzing the Asymmetric Effect of Renewable Energy Consumption on Environment in STIRPAT-Kaya-EKC Framework: A NARDL Approach for China” to improve the quality of our manuscript. We have reviewed the comments and generally agree with those. Accordingly, we have made changes to the paper that are in line with the editor’s observations to the extent practicable. We take this opportunity to thank each of the individuals involved in the process. We express our sincere gratitude for their in-depth reviews, which have helped significantly improve the paper's quality.
Please find below our response to each of the comments in the order in which they have been raised.
Reviewer # 2
Most of the comments have been addressed.
The comment about the trend treatment has been misinterpreted. The questions on hand is about the need for "trend adjustment" in the models. The authors have only plotted the trend of each series, without any reflections on what it is observed and how it may affect the analysis. This is not useful thus it is recommended to remove Graph 1, unless they reflect on the impact of the trend on the models. Is a trend adjustment required for these models?
Reply: Thanks for the encouraging reply on addressing the comments. However, The model's parameters in cumulative sum and cumulative sum of squares are entrenched inside the 5% critical lines. Thus, we may conclude that our model is perfectly stable even without considering trend adjustment. Therefore, we do not include it in our empirical model. However, we have removed Graph 1.
